

# Using crowdsourced web content for informing water systems operations in snow-dominated catchments

Matteo Giuliani[1], Andrea Castelletti[1,2], Roman Fedorov[1], and Piero Fraternali[1]

[1]Department of Electronics, Information and Bioengineering, Politecnico di Milano, Piazza L. da Vinci, 32, I-20133 Milano, Italy
[2]Institute of Environmental Engineering, ETH Zurich, Wolfgang-Pauli-Str. 15, CH-8093 Zurich, Switzerland

**Abstract.** Snow is a key component of the hydrologic cycle in many regions of the world. Despite recent advances in environmental monitoring are making a wide range of data available, continuous snow monitoring systems able to collect data at high spatial and temporal resolution are not well established yet, especially in inaccessible high latitude or mountainous regions. The unprecedented availability of user generated data on the Web is opening new opportunities for enhancing real-time monitoring and modeling of environmental systems based on data that are public, low-cost, and spatio-temporally dense. In this paper, we contribute a novel crowdsourcing procedure for extracting snow-related information from public web images, either produced by users or generated by touristic web cams. A fully automated process fetches mountain images from multiple sources, identifies the peaks present therein, and estimates virtual snow indexes representing a proxy of the snow covered area. Our procedure has the potential for complementing traditional snow-related information, minimizing costs and efforts for obtaining the virtual snow indexes and, at the same time, maximizing the portability of the procedure to several locations where such public images are available. The operational value of the obtained virtual snow indexes is assessed for a real world water management problem, the regulation of Lake Como, where we use these indexes for informing the daily operations of the lake. Numerical results show that such information is effective in extending the anticipation capacity of the lake operations, ultimately improving the system performance.

## 1 Introduction

Snow accumulation and melting are fundamental components of the hydrological cycle in many watersheds across the world (e.g., Mote et al., 2005; Holko et al., 2011). Approximately 40-50% of the Northern Hemisphere is covered by snow (Pepe et al., 2005) and snow plays a key role in mountain areas, which, in Europe, account for 40% of the total surface (Schuler et al., 2004).

In such contexts, an accurate characterization of snow availability and its evolution in time can be extremely valuable for a variety of operational purposes, from avalanche prediction (e.g., Perona et al., 2012; Schweizer et al., 2009), water systems operations through medium to long-term streamflow forecast (e.g., Wood and Lettenmaier, 2006; Anghileri et al., 2016), or drought risk management (e.g., Staudinger et al., 2014). The projected temperature increase induced by climate change, with





consequent reductions of large volumes of snowpack and acceleration of the water cycle in many mountainous areas, will further amplify the importance of better understanding snow dynamics (Barnett et al., 2005; Kunkel et al., 2016).

Snow processes are generally monitored through both ground monitoring networks (e.g., Brown and Braaten, 1998; López-Moreno and Nogués-Bravo, 2006) and remote sensing (for a review, see König et al., 2001; Dietz et al., 2012, and references

therein). Yet, both sources have serious limitations in alpine contexts mainly related to the high spatial (e.g., Newald and Lehning, 2011) and temporal variability of snow related processes (Blöschl, 1999; Egli, 2008; Gleason et al., 2016). Ground stations are generally very coarsely distributed. Satellite products provide data on a denser grid but are diversely constrained depending on the sensors installed (Muñoz et al., 2013). High spatial and temporal resolution imagery (i.e., daily maps with spatial resolution of about 500 m) can be derived from Moderate Resolution Imaging Spectroradiometer (MODIS) products,

which are, however, strongly affected by the weather because optical sensors cannot see the earth surface when clouds are present (Parajka and Blöschl, 2008). Space-board passive microwave radiometers (e.g., AMSR-E) penetrate clouds and provide accurate snow cover estimation, but have coarse spatial resolution (25 km). Finally, the use of active microwave systems (e.g. RADARSAT) is so far limited to the detection of liquid water content.

The last few years have seen a rising interest in complementing traditional observations by using cameras and short-range

visual content analysis techniques (Bradley and Clarke, 2011), which allow improving the temporal and spatial resolutions for specific applications. Many case studies showed that the use of one or several time-lapse cameras allows mapping both the spatial and temporal patterns of a variety of snow characteristics, including glacier velocity, snow cover changes, or detailed monitoring of snowfall interception (see Parajka et al., 2012, and references therein). However, most of these systems generally rely on cameras designed and positioned ad hoc (e.g., Hinkler et al., 2002), possibly including in the camera view some specific

objects, such as flags or sticks, which simplifies the calibration of geometry and colors (e.g., Floyd and Weiler, 2008; Laffly et al., 2012; Garvelmann et al., 2013). In addition, the use of these cameras is generally very expensive and often requires intensive manual efforts in the image processing phase. This latter includes a variety of crucial, time-consuming operations, such as the selection of photographs with good meteorological and visibility conditions, the photo-to-terrain alignment and orientation, and the labeling of snow covered pixels for estimating the total snow cover (e.g., DeBeer and Pomeroy, 2009;

Farinotti et al., 2010).

The availability on the web of large volumes of public, low-cost, and spatio-temporally dense data raises the question of whether it is possible to use such data as a supplement, or at least as a complement, to traditional monitoring systems in operational contexts. The main advantage of such public data, albeit collected for completely different purposes and with much lower quality standards, is that they can significantly increase the spatial and temporal coverage at little/no cost (Jacobs

et al., 2009; Graham et al., 2010). This idea is part of a growing application of so called "citizen science" approaches to water resources systems operation (Buytaert et al., 2014) and, more generally, to diverse environmental problems (Fraternali et al., 2012). Crowdsourced observations may act as low-cost virtual sensors in a variety of environmental contexts (Lowry and Fienen, 2013), for example, contributing to monitoring the dynamics of forests (e.g., Daume et al., 2014), storms (e.g., Good et al., 2014), or streamflow (e.g., Michelsen et al., 2016), with potential benefit in terms of the prediction of flood events and of

the timely delivery of alarms (e.g., Smith et al., 2015; Mazzoleni et al., 2015a, b; Fohringer et al., 2015; Le Boursicaud et al.,



2016). However, despite this interest in environmental public web and user generated data (Vitolo et al., 2015), most works focus on data collection and analysis, with limited assessment of the practical value of such crowdsourced information.

In this paper, we explore the potential for web and crowdsourced data to retrieve relevant information on snow availability and dynamics in a river basin, and assess the utility of such information in informing a real world decision making problem.
More precisely, we contribute a novel crowdsourcing procedure for extracting snow-related information from public web images, either produced by users or generated by touristic web cams, and we quantify the operational value of this information compared to other more traditional snow information, such as ground observations and a hybrid mix of satellite retrieved information, ground data, and model outputs. Our procedure employs an articulated architecture (Fedorov et al., 2015), which automatically crawls content from multiple web data sources with a content acquisition pipeline integrating public web cams
and user-generated photographs posted on Flickr. Next, the procedure retains only geo-tagged images containing a mountain skyline with high probability and identifies the visible mountain peaks in each image, using a digital elevation model (DEM). Then, a supervised learning classifier extracts a snow mask from each image, which distinguishes the image pixels as snow or no-snow. Finally, the resulting snow masks are post-processed to derive time series of virtual snow indexes (VSI) representing a proxy of the snow covered area.

The extracted VSI are used to inform water system operations. The evaluation is performed in the snow-dominated catchment of Lake Como, a regulated lake in Northern Italy, where snow melt is the most important contribution to the seasonal storage. The VSI operational value is quantified by comparing, via simulation, the performance of the lake operating policies designed using crowdsourced and traditional snow information, with the performance of the baseline policy obtained by regulating the lake without snow information (Giuliani et al., 2015). This form of assessment provides an indirect validation of the utility
of web and crowdsourced information. This is the only viable evaluation method, because the VSI that can be extracted from general-purpose mountain images and the traditional observational data collected with dedicated tools are not comparable directly due to the difference in their physical interpretation and spatio-temporal resolution (e.g., geo-located photos allow estimating the presence of snow, but not the physical measures usually employed in snow process models, such as the snow water equivalent).

The paper is organized as follows: in the next section, we introduce our methodology for the computation of VSI based on public web content and the assessment of their operational value. Section 3 describes the Lake Como study site, followed by the discussion of the numerical results. The last section concludes with final remarks and directions for further research.

## 2 Methods and tools

This section describes the methodology adopted in this work, which is illustrated in Figure 1. Details about each phase of the procedure are provided in the following sub-sections.




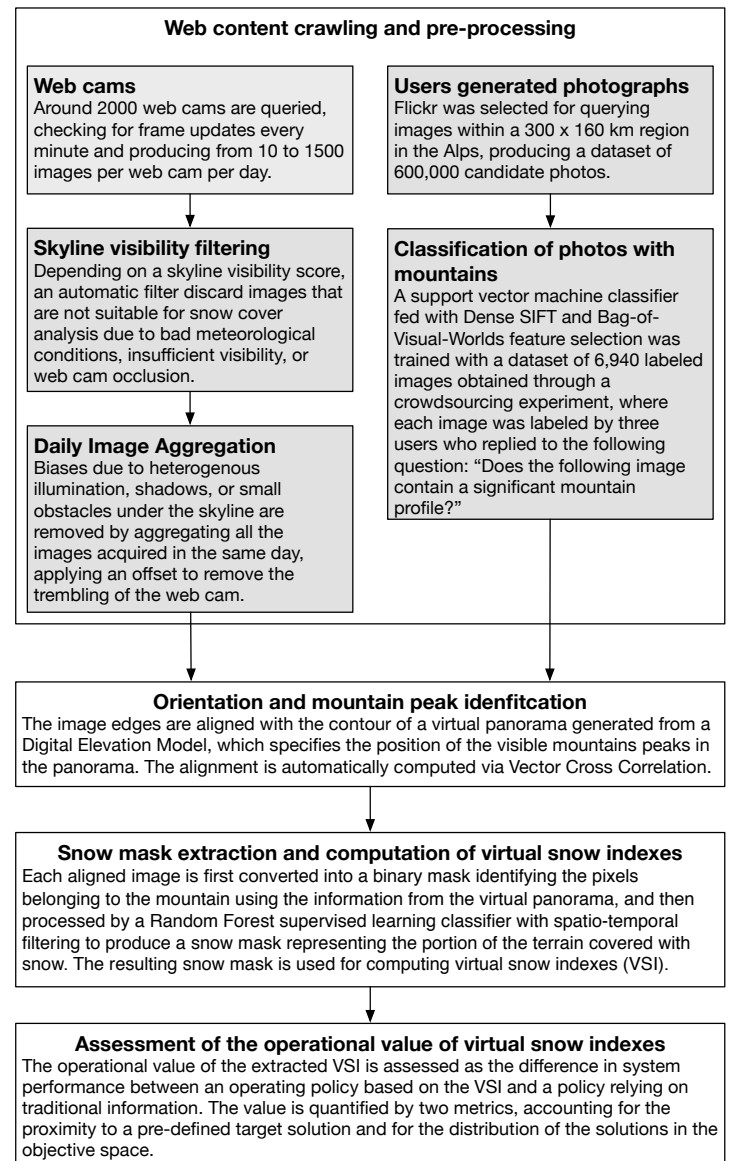

**Figure 1.** Flowchart of the methodology adopted in this study.

## 2.1    Web content crawling and pre-processing

Two types of public web content are considered, namely touristic web cams and mountains photographs from Flickr. In particular, web cams produce a temporally dense series of images of the same view, while crowdsourced photos have better spatial distributions but lower time coverage.





### 2.1.1 Public web cams

A web cam is a standalone camera positioned at a fixed known location, usually with a fixed orientation, which captures frames with a certain frequency and exposes them via a web service. Differently from surveillance web cams, which can provide real time updates (several frames per second, resulting basically in a video stream), public web cams deployed for touristic, meteorological, and publicity reasons update the current frame with lower frequency, typically from one minute to one hour. The public web cam processing phase consists of three main steps:

1. **Web cam image crawling**: public web cams most often expose a single fixed URL for the current frame, and change the image itself over time. This method simplifies the crawling, which amounts to checking the URL of the web cam periodically and downloading the image, when it changes with respect to the last acquisition. We collected the address of more than 3,500 web cams in the European Alps and manually inspected them, discarding those that do not frame a significant mountain profile, retaining nearly 2,000 web cams. Since December 2014, a crawler acquires all the images
of these web cams, checking for frame updates every minute, thus obtaining from 10 to 1,500 images per web cam per day, depending on the update frequency.

2. **Skyline Visibility Filtering**: web cams are crawled independently of the weather conditions. As a consequence, although the temporal density of web cam images guarantees a high number of input frames, filtering must be applied to discard unsuitable images that may bias the VSI computation (e.g., an image of a mountain covered by fog can be considered as completely covered by snow in the next steps). A random sampling of 1,000 images from 4 web cams in our data
set revealed that 67% of the images were not suitable for snow cover analysis due to adverse weather conditions (e.g., fog, heavy snowfall, or rain), insufficient visibility, or presence of mobile obstacles such as cars or persons. Therefore, the implemented filter automatically discards unsuitable images, identified by checking for occlusions of the mountain skyline. In practice, for each web cam, the pixels that belong to the skyline $\mathcal{L}$ are first identified manually on a sample
frame, and the binary skyline neighborhood mask $L$, which identifies pixels $p = (x, y)$ close to the skyline, is determined as follows:

$$L(p) = \begin{cases} 1 & \text{if } \exists p' \in \mathcal{L} : \|p - p'\| \leq \tau \\ 0 & \text{otherwise} \end{cases}, \tag{1}$$

where $\|\cdot\|$ is the Euclidean norm. In other words, $L$ is a binary mask of the same dimension as the web cam image containing a dilated skyline profile.

Then, for each web cam image, its binary edge map $E$ is computed, where a pixel is marked as an edge when it corresponds to an abrupt color variation. The binary matrix $E \odot L$, where $\odot$ denotes the pixel-wise product between two



images of the same size, represents the edges of the image that belong to the skyline. To check for occlusions, we compute a skyline visibility score $v$ defined as

$$v = f(E \odot L) \,/\, f(L) \tag{2}$$

where $f(\cdot)$ is a function that, given an image, returns the number of columns containing at least one non-zero entry. The value of $v$ ranges between 0 and 1, and can be intuitively seen as the percentage of the skyline which is visible in the given image. After set-up trials, we discard images with $v < \bar{v}$, where $\bar{v}$ is a fixed threshold equal to 0.75. The experimental validation of the filtering algorithm on 1,000 manually annotated images (i.e., frames manually classified as "good visibility" or "bad visibility") showed that the algorithm achieves a True Positive Rate (TPR) equal to 87.4%, while having False Positive Rate (FPR) equal to 3.5%.

3. **Daily Image Aggregation**: the images selected by the skyline visibility filter can still present several undesirable features due to shadows, solar glare, or temporary obstacles below the skyline (e.g., people standing in front of the camera). To attenuate such biases, assuming that the snow cover does not vary during a day significantly, we produce a single image for each web cam per day by aggregating all the images acquired in a same day. Such a Daily Median Image (DMI) is obtained as the median of every pixel across all the daily images accepted by the filter. Given a daily sample of $N$ images $I_1, \ldots, I_N$, the DMI is formally defined as

$$DMI(x,y) = med\{I_1(x,y), I_2(x,y), \ldots, I_N(x,y)\} \tag{3}$$

where $med\{\cdot\}$ is the median operator applied to the image pixel values. Figure 2 shows a DMI obtained from 11 daily images: it attenuates transient light conditions and removes the people standing in front of the web cam.

A second challenging aspect of DMI creation is the presence of web cam trembling (Latecki et al., 2005). The web cam orientation is not perfectly constant in time but may change slightly, especially in windy regions and when web cams are fixed to poles. To overcome this problem, we extract edge maps of all daily images and calculate through cross correlation the best offset of every image with respect to the reference represented by the first image of the day. The DMI is then determined from images normalized with such offset. Intuitively, this procedure can be seen as applying a small displacement to each image in order to obtain the best possible overlap between its edges and the edges of the first image of the day.

### 2.1.2 User generated photographs

The second source of mountain images are the photographs generated by common people and publicly available on social networks and photo sharing platforms. Although the volume of user generated photographs can not obviously reach the number





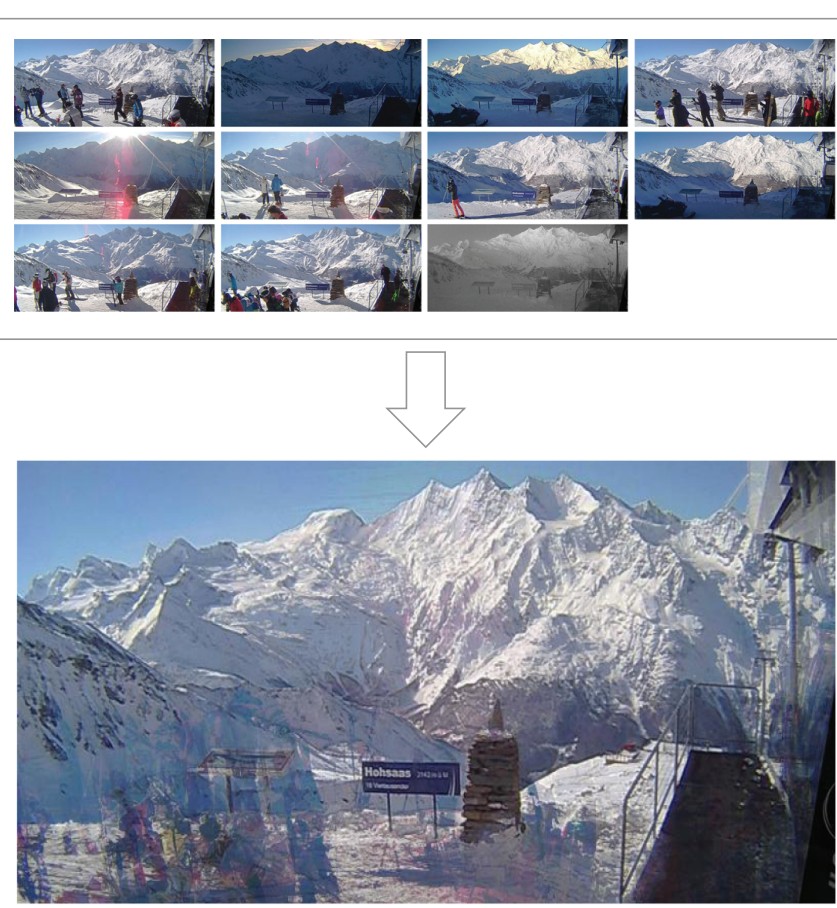

**Figure 2.** Example of Daily Median Image obtained from 11 images acquired in the same day.

of web cam images, user-generated photographs present higher spatial density. The web cams are indeed located in a few fixed locations, whereas the user photographs can be potentially acquired in any place.

We selected Flickr as the content source because it contains a high number of public photographs with associated geo-

20  tag (e.g., Serdyukov et al., 2009). Furthermore, Flickr does not remove the EXIF information present in the original images (Tesic, 2005); the EXIF container specifies several photo-related details, in particular the GPS location, the camera model and manufacturer, and optical information, such as the focal length used during the shot. This information is fundamental for the peak detection algorithm (see Section 2.2). A continuous search system was set up for querying images within a $300 \times 160$ km region in the Alps. At present, the Flickr search pipeline examined around 600,000 candidate photos. Differently

25  from registered web cams, which produce thousands of images of the same view, the user-generated photograph are taken at unknown location and may have an irrelevant content, and thus must be classified as relevant one-by-one. To this end, a supervised content-based classifier was developed to perform mountain image detection. The classifier was trained on a set





of 6,940 images randomly sampled from the very large crawled data set; the ground-truth images were classified manually through a crowdsourcing experiment. For each image, three users were presented with an image and asked to reply to the following question: "Does the image contain a significant mountain profile?". A web interface proposed a tutorial on how to annotate an image as positive (mountain image) or negative (non-mountain image). The experiment was conducted using an internal (non paid) crowd, collecting a total of more than 20,000 image classification labels. The aggregated ground-truth label of each image was then derived via majority voting. Approximately 23% of the original 6,940 images were classified as positive.

The automatic classification was performed with a Support Vector Machine (SVM) classifier fed with Dense SIFT and Bag-of-Visual-Worlds (BoVW) feature selectors (Fei-Fei and Perona, 2005). This technique relies on the idea that every image is composed by small patches (i.e., image portions), which somehow share common features with the images in the same class (i.e., images that do contain or do not contain mountains). Since the number of possible patches to observe is very large, the patches are split into a finite number of clusters. Each patch represents a visual word, which contributes to defining the content of the image. All the visual words of the image are aggregated into a histogram, which is then used as feature vector for the SVM classifier. To create a balanced data set, we retained all the positive samples and randomly selected the same number of negative samples. Then, we used around 70% of these images for training and validation, and the remaining 30% for testing. The performance attained by the classifier on the test data set is: 95.1% accuracy, 94.0% precision, and 96.3% recall.

### 2.2 Orientation and mountain peak identification

The orientation and mountain peak identification procedure (see Figure 3) is applied to the user generated photographs classified as positive and to the median daily images of web cams. In fact, although web cams are geo-located, the information regarding the orientation of the web cam and, consequently, the corresponding mountain peaks observed, is not available. In both cases, image orientation is estimated through the alignment with respect to a 360° virtual panorama generated using a digital elevation model (DEM) that specifies the position of the visible mountain peaks in the panorama.

The automatic alignment of an image to the virtual panorama requires scaling the image to achieve the same angular/pixel dimension. This step is performed by computing the image Field Of View (FOV), namely the size of the angle comprising the view. The FOV can be estimated from the image EXIF information, such as focal length, camera model, and manufacturer. Then, the procedure extracts the edge maps for both the scaled image and the virtual panorama. In particular, a skyline detection algorithm is applied to the edge maps to eliminate all the edges above the skyline (clouds and obstacles) and reducing gradually the strength of the edges below the skyline. Then, the best overlapping position between the image and the virtual panorama is identified with a Vector Cross Correlation (VCC) procedure (Baboud et al., 2011). The VCC finds the best horizontal overlap position of the image with respect to the panorama by maximizing the cross correlation score of the edges, also considering the estimated image orientation. The identified overlap position allows projecting the peak positions from the panorama to the image to estimate which peaks are visible and their coordinates in the image. When the image does not contain the EXIF information, the automatic orientation and mountain peak identification procedure can not be applied and the image requires a





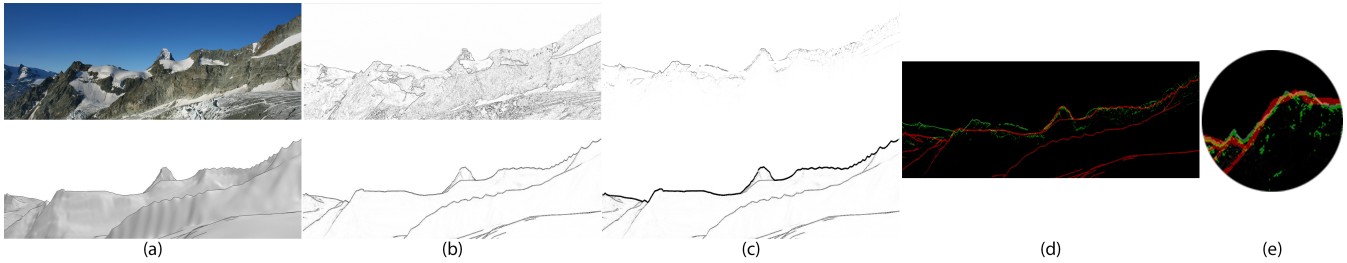

(a)    (b)    (c)    (d)    (e)

**Figure 3.** Example of the orientation and mountain peak identification procedure: (a) input image (top) and corresponding panorama (bottom); (b) edge maps; (c) skyline detection; (d) global alignment; (e) local alignment.

manual alignment with respect to the panorama. Finally, a local refinement of the alignment is obtained by repeating the VCC procedure for each peak and adjusting its position through the identification of the best match in its neighborhood region.

The orientation and peak identification algorithm was tested on a data set of 162 images randomly sampled from the web and manually aligned to the corresponding virtual panoramas to create the ground-truth data. Considering a tolerance of 3 deg, 75% of the image orientations were correctly estimated. The accuracy grows to 77.6% for photos with no clouds and to 81.6% for photos with no mountain slopes in the very short range (the effect of GPS errors is more sensible if mountains are close to the shooting location). The average peak positioning error resulted to be 0.78 deg.

### 2.3 Snow mask extraction and computation of virtual snow indexes

The third step of the procedure is the conversion of the snow information contained in the aligned image into one or more VSI associated to the mountain viewpoint portrayed in the photo. This phase requires estimating a snow mask representing the portion of the terrain that is covered with snow. Formally, let $I$ denote an image and $M$ a binary mask having the same size as $I$, where $M(x,y) = 1$ indicates that the pixel $p(x,y)$ of the image belongs to the mountain area, or $M(x,y) = 0$ otherwise. The binary mask $M$ is derived from the alignment of the image with the virtual panorama (see the previous section), which allows distinguishing pixels that correspond to terrain or sky.

The pair $(I, M)$ is processed by a pixel-level binary classifier, which extracts the snow mask $S$ by assigning to each pixel a label denoting the presence of snow ($S(x,y) = K_1$), non-snow ($S(x,y) = K_2$), and sky ($S(x,y) = K_3$) as shown in Figure 4. We computed snow masks using the Random Forest supervised learning classifier with spatio-temporal median smoothing of the output (Liaw and Wiener, 2002). Such classifier discriminates the presence of snow in a pixel based on its color and on the color of the neighbor pixels. Moreover, it applies a spatio-temporal median filter to smooth the snow variation and attenuate the errors. Smoothing implements the assumption that pixels close to each other in the same image and pixels in the same position in images close in time should belong to the same class (i.e., snow/non-snow). The training and testing of the classifier was performed on a data set including 59 annotated images, containing more than 7 million single pixel ground-truth labels. The accuracy attained by the classifier is 93.5%, outperforming other existing methods for pixel-level classification of snow presence (Fedorov et al., 2015).





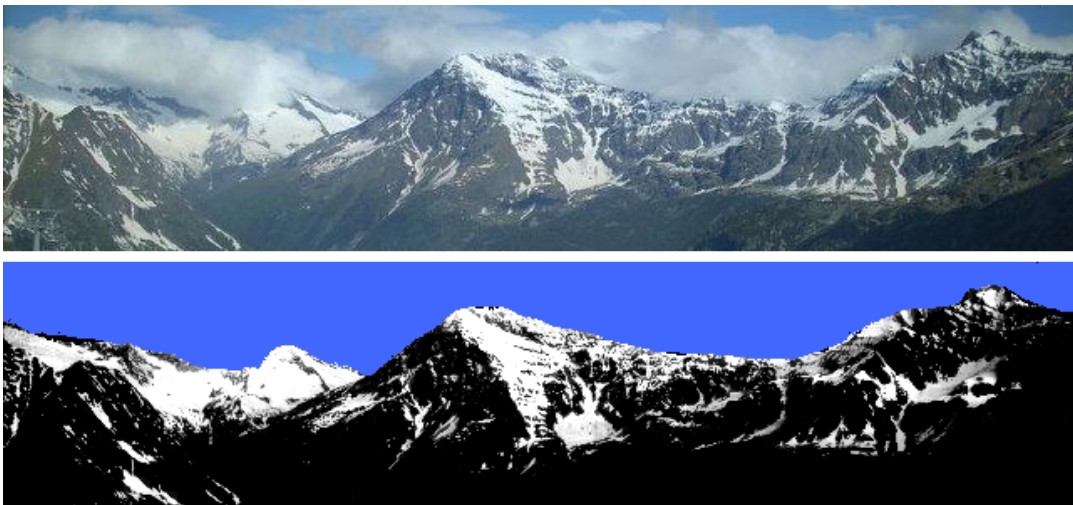

**Figure 4.** Example of an image (top) and the computed snow mask (bottom), where white stands for snow, black for non-snow, and blue for sky.

Finally, different VSI are computed from the snow masks $S$, potentially considering also the altitude associated to each pixel, which can be determined from the image to virtual panorama alignment. In this work, we report the results obtained with a Virtual Snow Index $\sigma$ representing a proxy of the snow cover area, defined as follows:

$$\sigma = \sum_{p(x,y) \in I} \Phi(p(x,y)) \quad \text{where}$$

$$\Phi(p(x,y)) = \begin{cases} 1 & \text{if } S(x,y) = K_1 \\ 0 & \text{otherwise} \end{cases} \tag{4}$$

## 2.4 Assessment of the operational value of virtual snow indexes

The operational value of the extracted VSI is assessed as the difference in system performance between an operating policy based upon the VSI and a policy relying on more traditional information, including water availability in the lake and day of the year. In particular, the operating policies are computed by solving a multi-objective optimal control problem (Castelletti et al., 2008) formulated as follows:

$$p^* = \arg\min_p \mathbf{J} = |J^1, \dots, J^q| \tag{5}$$

where the policy $p$ is defined as a closed loop control policy that determines the release decision $u_t = p(d_t, \mathbf{x}_t, \mathcal{I}_t)$ at each time step $t$ as dependent on the day of the year $d_t$, the current state of the system $\mathbf{x}_t$ (i.e., the level of the lake at time $t$), and a vector

(c) Author(s) 2016. CC-BY 3.0 License.





25  of exogenous information $\mathcal{I}_t$ (i.e., variables that are observed but are not endogenous in the problem formulation and hence are not modeled). Note that the resolution of Problem (5) does not yield a unique optimal solution but a set of Pareto optimal solutions.

The most common technique to solve Problem (5) is Dynamic Programming (Bellman, 1957). However, DP is severely limited by the curse of modeling in designing operating policies conditioned on exogenous information (Tsitsiklis and Van Roy, 1996) and by the curse of multiple objectives in exploring multidimensional tradeoffs (Powell, 2007). We therefore solve Problem (5) by means of Evolutionary Multi-Objective Direct Policy Search (Giuliani et al., 2016), an approximate dynamic

programming approach that combines direct policy search, nonlinear approximating networks, and multi-objective evolutionary algorithms. EMODPS allows the direct use of exogenous information through a partially data-driven controller tuning approach (Formentin et al., 2013), where the operating policy is defined as a nonlinear approximating network directly conditioned on observation of exogenous information. However, the selected policy parameterization strongly influences the selection of the optimization approach as the number of parameters necessary to obtain a good approximation for the unknown optimal control

policy grows with the increasing dimension of the policy's argument (Zoppoli et al., 2002). Since the optimization of the policy parameters requires searching high dimensional spaces that map to stochastic and multimodal objective function values, global optimization methods such as evolutionary algorithms are preferred to gradient-based methods (Heidrich-Meisner and Igel, 2008).

Given the Pareto optimal solutions of Problem (5), the operational value of the estimated VSI is quantified by means of two

metrics (Giuliani et al., 2015). The first metric is a measure of the proximity between a pre-defined target solution $\mathbf{J}_T$ and the closest alternative in the Pareto front of the policy under examination, i.e.

$$D_{min} = \min_{i=1,...,N} \|\mathbf{J}_T - \mathbf{J}_i\| \tag{6}$$

where $\|\cdot\|$ stands for the (normalized) Euclidean norm, $N$ is the number of solutions in the Pareto front under exam, and $\mathbf{J}_i$ is the performance of the $i$-th solution in the Pareto front. The lower $D_{min}$, the closer to the target the performance.

A more informative assessment can be done by evaluating not only how close a given policy can get to a pre-defined target solution but, more generally, how the Pareto approximate solutions distribute in the objectives space. Among the commonly used metrics adopted in the literature (see Maier et al. (2014) and references therein), we adopt the hypervolume indicator ($HV$), which captures both the convergence of the Pareto front under examination $\mathcal{F}$ to the optimal one $\mathcal{F}^*$ as well as the representation of the full extent of tradeoffs in the objective space. The hypervolume measures the volume of objective space





dominated ($\preceq$) by the considered set of solutions. This metrics allows set-to-set evaluations, where the Pareto Front with higher $HV$ is considered better. $HV$ is calculated as the hypervolume ratio between $\mathcal{F}$ and $\mathcal{F}^*$, formally defined as:

$$HV(\mathcal{F}, \mathcal{F}^*) = \frac{\int \alpha_{\mathcal{F}}(\mathbf{x})dx}{\int \alpha_{\mathcal{F}^*}(\mathbf{x})dx} \quad \text{where}$$

$$\alpha_{\mathcal{F}}(\mathbf{x}) = \begin{cases} 1 & \text{if } \exists \mathbf{x}' \in \mathcal{F} \text{ such that } \mathbf{x}' \preceq \mathbf{x} \\ 0 & \text{otherwise} \end{cases} \tag{7}$$

## 3 Lake Como study site

### 3.1 System description

Lake Como is a regulated lake in the Adda River basin, Italy (Figure 5). The lake has an active storage capacity of 254 Mm$^3$
and is fed by a 3,500 km$^2$ alpine catchment that reaches altitudes over 4,000 m asl. Downstream from the lake, the Adda River serves a dense network of irrigation canals belonging to four agricultural districts for a total irrigated area of 1,400 km$^2$ (green area in Figure 5). Major cultivated crops are maize and temporary grasslands, while minor crops include rice, soybean, wheat, tomato, and barley. The hydro-meteorological regime in the catchment is the typical sub-alpine one, with scarce discharge in winter and summer, and peaks in late spring and autumn due to snowmelt and rainfall, respectively. In particular, snowmelt
from May to July is the most important contribution to the formation of the seasonal storage (Figure 6).

The alpine orography constrains the accurate monitoring of snow dynamics. The existing ground stations (46 over the 10,500 km$^2$ alpine area in the Lombardy region) provide a very coarse coverage of the region and are not sufficient to reliably monitor the snow coverage and the associated water content. This is instead estimated by the Regional Agency for Environmental Protection (Agenzia Regionale per la Protezione dell'Ambiente - ARPA), which produces estimates of snow water equivalent
(SWE) through a hybrid procedure combining snow height and temperature data from ground stations, measures of snow density in few specific locations, satellite retrieved data of snow cover from MODIS, and model outputs for spatially interpolating these data. As a result of this complex procedure, ARPA elaborates a weekly estimate of SWE. Such reports are delivered only weekly due to the well known limitations of snow products derived from optical sensors associated to the frequent satellite occlusion by cloud coverage. This limitation is particularly restrictive in the alpine region, where previous studies observed an average cloud occlusion of 63% over a five year monitoring period (Parajka and Blöschl, 2006), with critical episodes of cloud coverage lasting for more than 25 days per month in winter time. On the contrary, web cams are less affected by cloud coverage and can provide observations during cloudy days as shown illustratively in Figure 7. In this study, we contrast the
5 operational value in informing the lake operation of three different snow-related data sources: ($i$) daily observations of snow height from coarsely distributed ground stations; ($ii$) weekly SWE estimate provided by ARPA; ($iii$) daily values of the VSI $\sigma$ extracted from public web images.



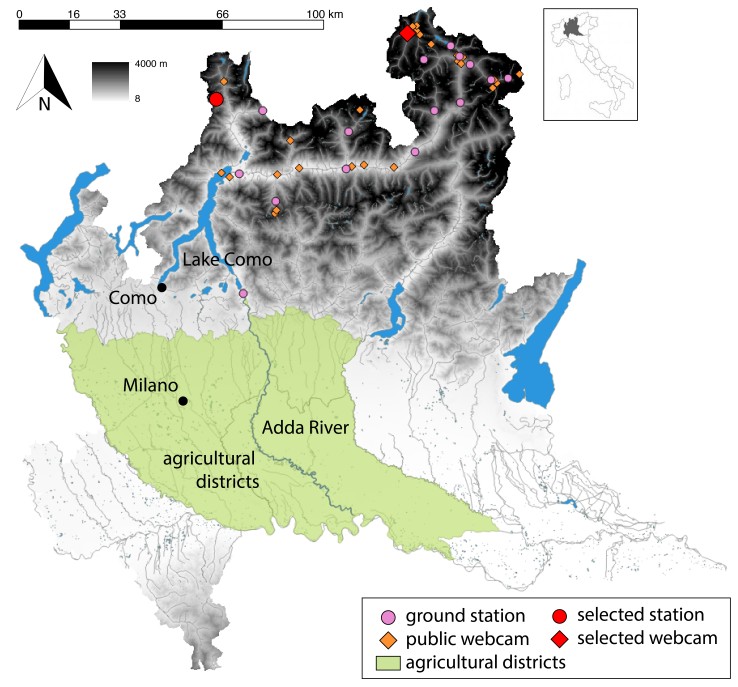

**Figure 5.** Adda River Basin: Lake Como, Adda River, downstream agricultural districts, ground stations, and public web cams.

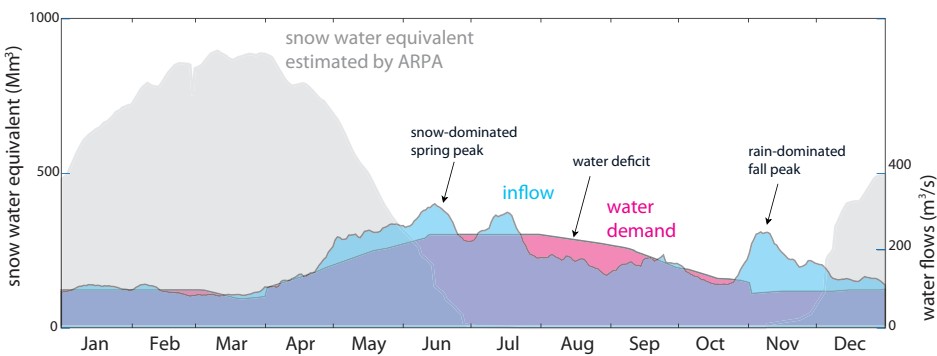

**Figure 6.** Hydro-meteorological regime of Lake Como.

The existing regulation of the lake is driven by two primary, competing objectives: water supply, mainly for irrigation, and flood control in the city of Como, which is the lowest point of the lake shoreline. In particular, the agricultural districts downstream would like to store the snowmelt volume for the summer water demand peak, when the natural inflow is not sufficient to satisfy the irrigation requirements (see the magenta area in Figure 6). Yet, storing such water increases the lake level and, consequently, the flood risk, which would be instead minimized by keeping the lake level as low as possible. On the





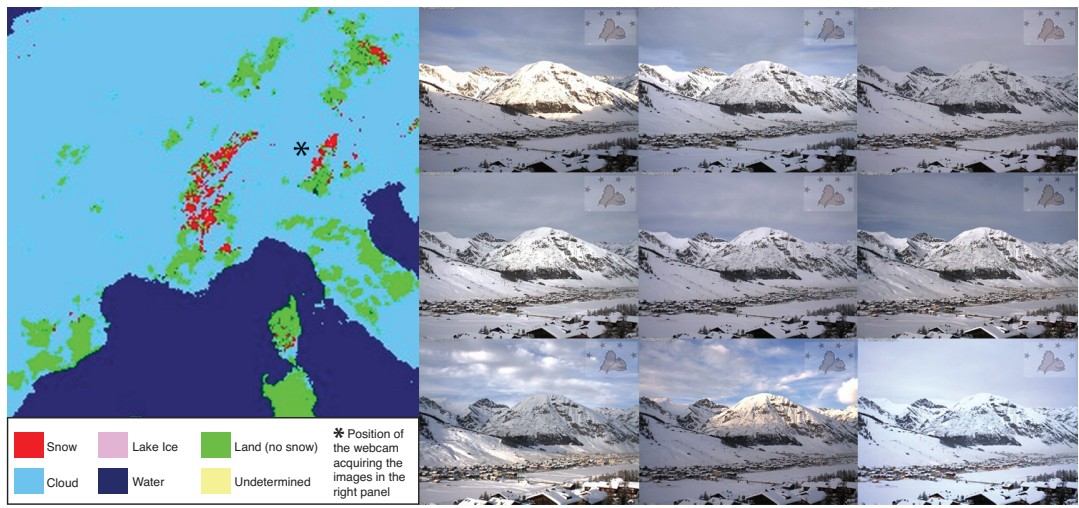

**Figure 7.** Comparison of MODIS daily snow cover map (left panel) with the images acquired by a web cam (right panel) on Jan. 9, 2014 at the location denoted by the asterisk in the map.

basis of previous works (e.g., Castelletti et al., 2010; Anghileri et al., 2011, 2013; Giuliani and Castelletti, 2016), these two objectives are formulated as follows:

15      – *Flood control*: the average annual number of flooding days in the evaluation horizon $H$, defined as days when the lake level $h_t$ is higher than the flooding threshold ($\bar{h}$=1.24 m):

$$J^{flood} = \frac{1}{H/365} \sum_{t=1}^{H} \Lambda(h_t) \quad \text{where}$$

$$\Lambda(h_t) = \begin{cases} 1 & \text{if } h_t > \bar{h} \\ 0 & \text{otherwise} \end{cases} \tag{8}$$

     – *Irrigation supply*: the daily average quadratic water deficit with respect to the daily water demand $w_t$ of the downstream system, subject to the minimum environmental flow constraint $q^{MEF}$ to ensure adequate environmental conditions in

20      the Adda River:

$$J^{irr} = \frac{1}{H} \sum_{t=1}^{H} \max\left(w_t - \max(r_{t+1} - q^{MEF}, 0), 0\right)^2 \tag{9}$$

This quadratic formulation aims to penalize severe deficits in a single time step, while allowing for more frequent, small shortages (Hashimoto et al., 1982).





## 3.2 Experiment setting

Our assessment of the operational value of the VSI relies on the comparison of the performance attained by informing the operating policies of Lake Como with alternative snow-related information: $(i)$ policies P1 informed by snow height observations from ground stations; $(ii)$ policies P2 informed by SWE estimates provided by ARPA; $(iii)$ policies P3 informed by the virtual snow index $\sigma$. Performance is evaluated against an upper bound solution, designed assuming perfect foresight of future inflows, and a baseline solution, corresponding to a traditional regulation conditioned on the day of the year and the lake level. The experimental setting is structured as follows:

– *Observational data*: we consider the time horizon 2013-2014 over which time series of snow height, SWE estimate, and VSI are available. In particular, snow height data are measured at the Truzzo ground station, while the VSI derive from the images of a web cam in Livigno (see Figure 5); both sources have time series covering the selected time horizon.

– *Informed solutions*: the operating policies P1, P2, and P3 are designed via EMODPS by parameterizing the policies as Gaussian radial basis functions, which have been demonstrated to be effective in solving this type of multi-objective policy design problems (Giuliani et al., 2014a, b), particularly when exogenous information is used for conditioning the operations (Giuliani et al., 2015). To perform the optimization, we use the self-adaptive Borg MOEA (Hadka and Reed, 2013), which has been shown to be highly robust in solving multi-objective optimal control problems, where it met or exceeded the performance of other state-of-the-art MOEAs (Zatarain-Salazar et al., 2016). Each optimization was run for 2 million function evaluations. To improve solution diversity and avoid dependence on randomness, the solution set from each formulation is the result of 30 random optimization trials. The final set of Pareto optimal policies for each experiment is defined as the set of non-dominated solutions from the results of all the optimization trials.

– *Upper bound solution*: this ideal set of operating policies, which assume perfect foresight of future inflows, were designed via Deterministic Dynamic Programming over the 2-years (2013-2014). The weighting method is used to aggregate the 2 operating objectives (i.e., flood control and irrigation) into a single objective, via convex combination.

– *Baseline solution*: the traditional regulation of the lake is represented in terms of a set of operating policies conditioned on the day of the year $d_t$ and on the lake level $h_t$. Also these policies were designed via EMODPS.

## 4 Results and discussion

A first qualitative analysis of the Virtual Snow Index $\sigma$ defined in eq. (4) can be performed by comparatively analyzing the trajectory of this VSI with respect to the snow height observations in the closest ground station (i.e., Oga San Colombano, located around 15 km far from the web cam) or with respect to some physical variables closely related to the snow dynamics. Figure 8 contrasts the historical trajectory of $\sigma$ in 2013 with the trajectories of snow height observations at Oga San Colombano station (left panel) and of the freezing level (right panel). Despite some differences due to the different locations of the web cam and the ground station, the first comparison shows similar temporal patterns: most of the snowmelt occurs between April





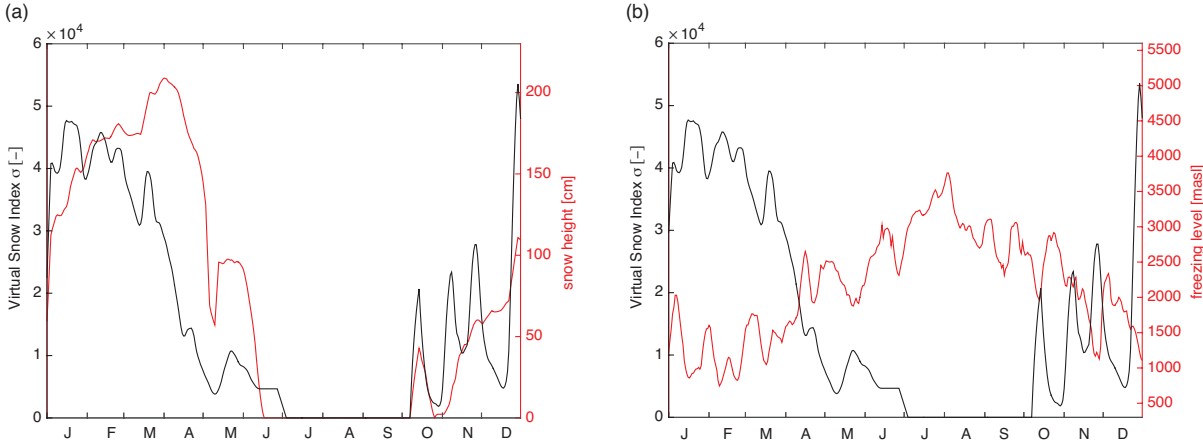

**Figure 8.** Comparison of the trajectories in 2013 of the Virtual Snow Index $\sigma$ with the snow height measured at Oga San Colombano (left panel) and with the freezing level (right panel).

and first half of May, followed by a late snowfall at the end of May; no snow is present since late June, with the first snowfall of the next winter observed in early October. The comparison between $\sigma$ and the freezing level shows a negative correlation between low values of freezing level from January to March as well as in November and December, which are associated to high values of $\sigma$. On the contrary, the freezing level increases in summer time in correspondence to low and zero values of $\sigma$. Moreover, it is worth noting the consistency in the oscillations of the two trajectories especially in winter time, when the snow accumulation is captured by increasing values of $\sigma$ associated to decreasing freezing levels and, viceversa, the snow melting corresponds to decreasing values of $\sigma$ and increasing freezing levels.

To further demonstrate the value of $\sigma$, we then quantified its operational value for informing the Lake Como operations (see Section 2.4). The performance of this set of informed operating policies (P3) is contrasted with the baseline solution, namely the traditional lake regulation conditioned on the day of the year and the lake level, and the upper bound solution, namely an ideal set of policies designed under the assumption of perfect foresight of future inflows. The same experiment is repeated using either ground observations of snow height (P1) or SWE data provided by the ARPA (P2) in order to validate the value of the VSI information with respect to traditional data sources.

Figure 9 illustrates the performance of the different set of solutions in terms of flood control ($J^{flood}$) and irrigation supply ($J^{irr}$), evaluated over the horizon 2013-2014. The arrows indicate the direction of increasing preference, with the best solution located in the bottom-left corner of the figure. Visual comparison of the baseline (blue circles) and upper bound solutions (black squares) shows the potential space for improvement generated by the ideal perfect information of the future inflows trajectories. A quantitative measure of this space is provided by the values of the two metrics $D_{min}$ and $HV$ introduced in Section 2.4. Table 1 shows that the normalized distance between the closest baseline solution to the target upper bound solution is 0.342, with this gap confirmed also for the entire set of solutions by the 0.292 difference in terms of hypervolume indicator. Valuable





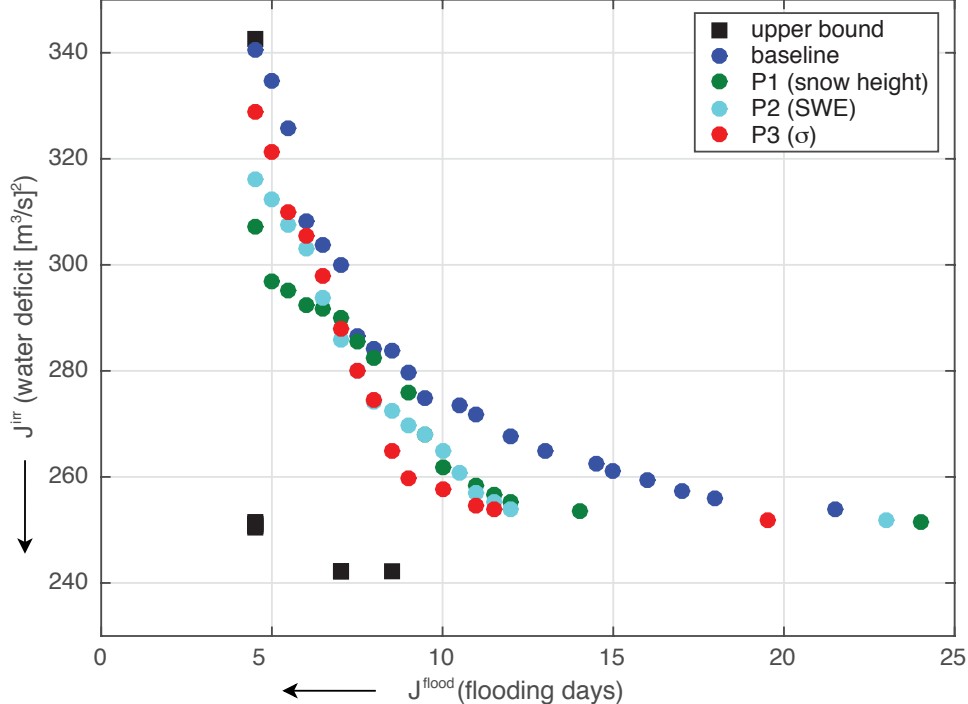

**Figure 9.** Performance obtained by different Lake Como operating policies informed with ground observations (P1 - green circles), SWE estimated by ARPA (P2 - cyan circles), or virtual snow indexes (P3 - red circles). The performance of these solutions is contrasted with the upper bound of the system performance (black squares) and the baseline operating policies (blue circles).

snow-related information is hence expected to fill the gap between the baseline and upper bound solutions. It is interesting to observe that, beside improving the performance of the operating policies with respect to both the objectives, the use of perfect information reduces the conflict between flood control and water supply, and discovers a number of solutions close to the independent optima of the two objectives, including the selected target solution $\mathbf{J}_T = (4.5; 250.6)$.

Given the references provided by the baseline and upper bound solutions, we can assess the operational value of different snow-related information by looking at the performance of informed operating policies, represented by the green, cyan, and red circles in Figure 9. Not surprisingly, numerical results show that enlarging the information used in the lake operations by accounting for the snow dynamics in the upstream catchment is producing an improvement of the system performance. In fact, the baseline solutions are completely dominated by the sets P1, P2, and P3. These informed operating policies successfully exploit the available snow data to implicitly obtain a medium to long term forecast of the future water availability due to snow melt, which supports the daily operations of the lake balancing flood protection on the short term and water supply on the long one. Overall, the three sets of Pareto optimal solutions, obtained using different snow information, attain similar performance, thus suggesting that the VSI can be considered equivalent to the other two physically based indexes. Figure 9 also shows that





policies P1 are the best for very low values of $J^{flood}$ but high values of $J^{irr}$, while policies P3 result to be the best in the compromise region of the objectives space (i.e., $J^{flood} < 10$ days and $J^{irr} < 275$ (m$^3$/s)$^2$), which is likely including the most interesting solutions for the lake operator as they successfully balance the system tradeoffs.

**Table 1.** Operational value of the VSI quantified by the two metrics introduced in Section 2.4.

| Policy | $D_{min}$ | $\Delta D_{min}$ | $HV$ | $\Delta HV$ |
|---|---|---|---|---|
| baseline | 0.342 | - | 0.708 | - |
| P1 (snow height) | 0.291 | 15.1% | 0.788 | 11.3% |
| P2 (SWE) | 0.290 | 15.2% | 0.785 | 10.9% |
| P3 ($\sigma$) | 0.238 | 30.4% | 0.790 | 11.6% |
| upper bound | 0.0 | - | 1.0 | - |

Finally, the values of the metrics reported in Table 1 confirm this visual evaluation. The three sets P1, P2, and P3 attain similar values of hypervolume indicator, which assesses the quality of the entire set of solutions. Interestingly, the policies P3 relying on the VSI outperform the other informed solutions both in terms of proximity to the target solution (i.e., lowest value of $D_{min}$) as well as quality of the entire Pareto front (i.e., highest value of $HV$). Although the differences in terms of hypervolume are limited, the operational value of $\sigma$ in terms of $D_{min}$ is relevant and improves the performance of the baseline solutions by 30%, doubling the improvement achievable by using either snow height or SWE data.

## 5 Conclusions

In this paper, we present a web content processing architecture for extracting snow-related information from public web images, either produced by users or generated by touristic web cams. The images, crawled from multiple web data sources, are automatically processed to derive time series of virtual snow indexes representing a proxy of the snow covered area. We then quantify the operational value of such data for informing the operations of Lake Como.

Numerical analysis shows that the time series of the virtual snow index extracted from a representative web cam is positively correlated with the snow height observations from ground stations and negatively correlated with the freezing level's dynamics. Moreover, our results demonstrate that the operational value of the virtual snow index meets or exceeds the one of traditional snow information. While the use of any snow information allows attaining a 10% increase in the hypervolume indicator with respect to the baseline system operations, the operating policies that use the virtual snow index are the closest to the target solution, selected as a good compromise between flood control and irrigation supply.

Future research efforts will focus on consolidating this approach by extending the evaluation horizon to better understand the system dynamics in terms of snow accumulation and melting as well as of the informed lake operations. In parallel, the amount of web content is expected to increase, potentially improving the spatial and temporal resolution of the generated snow-related information as well as its operational value. We have indeed developed a gamified web portal (http://snowwatch.polimi.it/)



where users can cooperatively access and enrich the data set of alpine mountain images. This portal is also expected to facilitate the users' engagement, fostering a more active participation to our image collection effort. Finally, our architecture can be employed in other snow-dominated catchments, such as Canada (Desreumaux et al., 2014), Lebanon (Mhawej et al., 2014), and Morocco (Marchane et al., 2015), as well as in different environmental problems that may benefit from using public web

content sources as low-cost virtual sensors, including sediment monitoring in river beds or vegetation monitoring in remote mountain regions.

*Acknowledgements.*  The authors would like to thank Agenzia Regionale per la Protezione dell'Ambiente, especially Dario Bellingeri and Enrico Zini, and Luigi Bertoli from Consorzio dell'Adda for providing the data used in this study. The work has been partially funded by the Proactive FESR project of Region Lombardy (grant n. 2760/2013) and by the IMPREX project funded by the European Commission under

the Horizon 2020 framework programme (grant n. 641811).





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
