# Peer review of "Using crowdsourced web content for informing water systems operations in snow-dominated catchments"

_Hydrology and Earth System Sciences, 2016_

## Referee Comment (RC1) · Anonymous Referee #1 · 3 Oct 2016

This is an interesting paper that ultimately I would like to see published in Hydrology and Earth System Sciences journal. This manuscript proposes a procedure for automatically extracting snow-related information from heterogeneous sources. I really enjoyed reading the paper, which deal with the important and timely issue of notable interest and modernity especially for the HESS readership. The paper accurately presents the methods and results. I have just a couple of minor comments/suggestions for the authors to consider.

From the introduction and methods sections, the authors mainly focused in the description of the approach used to derive VSI information from webcams and people pictures. However, in the results sections only a figure is reported to discuss the benefits of such

approach. I would like to see more analysis regarding the results achieved using the method reported in sections 2.1, 2.2 and 2.3. For example, it would be interesting to discuss issues in define skyline or to show a comparison between snow information extractions (of a certain point) using both web camera and user generated picture (if possible).

It is not clear to me how the information of VSI was used to estimate physical variable like ht in eq(10). Did the authors used any hydrological models? If yes, I think it would be appropriate to give a brief description in the methods section.

Finally, I believe authors should clearly define the limitations of this study (e.g. computational time of the imagine process and availability of public photo from people) in the conclusion section

---

## Referee Comment (RC2) · S. Gascoin (Referee) · 15 Oct 2016

This paper presents an approach to supplement in situ and satellite data in snow dominated watersheds by using publicly available webcam images and flickr photographs. The authors describe a complete procedure from the crawling of the images to the application of the extracted information on the regulation policy of a reservoir lake.

I enjoyed reading this paper and I concur with reviewer 1 that it deserves publication. I am also left with the feeling that the authors may have somehow eluded the limitations of their approach. The discussion should provide a more balanced analysis, e.g. by discussing the computation cost and data storage issues, the minimal amount or frequency of images to reach a stable solution in the VSI, and most importantly the steps

that require human intervention (see specific comments marked (A) and (B) below).

I spent some time to play around with this type of data[1] so I can imagine the tedious work and the challenges to automatically filter, align and classify webcams or photos. I encourage the authors to distribute an open source implementation of their processing to foster the development of similar applications in other regions.

I provided below a list of points that should be clarified. I hope that the authors will find my comments useful and look forward to reading an updated version. (NB. the line numbering of the manuscript is awkward, maybe an issue with the Copernicus LaTeX style file)

Specific comments:

P02-L12: AMSR-E derived SWE is generally not considered as "accurate" in mountain regions. Please modify or provide a reference to justify.

P03-L20: I disagree that the assessment of the VSI through the Lake Como experiment is the "only viable evaluation method". There are other validation approaches, including more direct approaches like a comparison with terrestrial time lapse cameras, comparison with high resolution satellite snow maps, etc. Please clarify or remove this sentence.

P05-L19: the skyline is manually defined for a first image. Do you mean that a skyline was manually digitalized on 2000 images (see P05-L09)? If yes this should be more clearly acknowledged. (A)

P05-Eq1: symbols p' and $\tau$ are not defined.

P05-L26: specify what is the edge detection algorithm.

P06-L09: why "cross" correlation? I would say correlation only.
* * *
[1]"Using kittens to unlock photo-sharing website datasets": http://www.cesbio.ups-tlse.fr/multitemp/?p=7317, which is citable as a publication of the Journal of Brief Ideas, doi:10.5281/zenodo.44809 (just saying...)

[Figure]

P06-L11: do you define a maximum offset to reduce the computation time, and if yes, how?

P08-L21: this is unclear to me: from the edge images, how do you extract the skyline? If this algorithm works, why was it not applied to the webcam images as well? I foresee many obstacles at this step, like the confusion of cloud edges or snow patches edges with skyline edges.

P09-L05: what does "local refinement" mean? do you mean a locally varying transformation of the image? If yes specify the method.

P09-L05 (sect 2.3): here I understand that you have used a supervised classification to get the snow mask. Then I suggest to explicit the number of samples and the method to define them. (B)

P12-L07 (at the end of the page...): please indicate the number of webcam images and the number of flickr photos that were used for this experiment.

P14-Eq9: define r.

P16-L32: did you try to use the freezing level as an input to the regulation model?

P18-L05: I created an account and logged in to this website to give it a try but the alignment tool was not really working. The page was not responding when I clicked "continue". It might be a browser issue (I used Firefox 49 on MacOS).

P19-L09: I am not convinced with the potential of this method in the Atlas mountains because there are few operating webcams and probably a much lower amount of wintertime public photos than in the Alps.

---

## Author Response (AR1)

Reply to reviewers about paper HESS-2016-400

**Using crowdsourced web content for informing water systems operations in snow-dominated catchments**

M. Giuliani, A. Castelletti, R. Fedorov, P. Fraternali

Matteo Giuliani
Department of Electronics, Information, and Bioengineering, Politecnico di Milano

Via Ponzio 34/5, 20133 Milano, Italy
Tel: +39 02 2399 9040
E-mail: matteo.giuliani@polimi.it

Dear Editor,

We would like to thank you and the reviewers for the helpful review.

We took all your points into consideration and revised the manuscript accordingly.

A detailed reply to reviewers is attached below. In preparing our response, all references to line numbers, equations, and figures are based on the revised manuscript; authors' replies as well as the changes tracked in the manuscript are in blue.

Sincerely,

Matteo Giuliani

**Referee comment #1**

This is an interesting paper that ultimately I would like to see published in Hydrology and Earth System Sciences journal. This manuscript proposes a procedure for automatically extracting snow-related information from heterogeneous sources. I really enjoyed reading the paper, which deal with the important and timely issue of notable interest and modernity especially for the HESS readership. The paper accurately presents the methods and results. I have just a couple of minor comments/suggestions for the authors to consider.
We thank the referee for the positive comment.

From the introduction and methods sections, the authors mainly focused in the description of the approach used to derive VSI information from webcams and people pictures. However, in the results sections only a figure is reported to discuss the benefits of such approach. I would like to see more analysis regarding the results achieved using the method reported in sections 2.1, 2.2 and 2.3. For example, it would be interesting to discuss issues in define skyline or to show a comparison between snow information extractions (of a certain point) using both web camera and user generated picture (if possible).
The results focus on quantifying the value of crowdsourced information in the operations of water systems, which is the main contribution of the paper. A detailed technical analysis of the image processing architecture is reported in Fedorov et al. (2015) including, for example, the comparison of the accuracy obtained with different feature extractors algorithms or the performance in the photo-to-terrain alignment (see tables below). To avoid replication, we did not include them in the paper and, following the reviewer suggestion, we added a sentence to direct the reader interested in the first part of the procedure toward the other paper (pag. 3, lines 29-30).
Finally, we agree with the reviewer that a direct comparison between the information extracted from both a webcam and a user-generated photo would be absolutely interesting. Unfortunately, at this stage we have not overlapping data to perform such comparison. We added this analysis as a possible future research, which, hopefully, will be possible thanks to the continuous acquisition of new web content through our portal (pag. 19, lines 13-14).

TABLE II: Results obtained by different feature extractors for the image classification problem (mountain vs. no-mountain).

| feature | $C$ | $\gamma$ | accuracy | precision | recall |
|---|---|---|---|---|---|
| Dense SIFT | 3.3 | 0.66 | **95.1** | **94.0** | **96.3** |
| HOG2x2 | 3.3 | 0.033 | 94.7 | 93.9 | 95.5 |
| SSIM | 0.66 | 0.33 | 93.0 | 92.5 | 93.5 |
| GIST | 0.33 | 1 | 87.61 | 82.64 | 95.21 |

TABLE IV: Performance results of the photo-to-terrain alignment algorithm (by dataset categories and photograph content properties)

| | $P_{1,1}^{G}$ | $P_{1,3}^{G}$ | $P_{1,3}^{R}$ |
|---|---|---|---|
| All images | 69.6% | 81.8% | 75.0% |
| Absence of clouds | 72.4% | 82.9% | 77.6% |
| Presence of clouds | 66.7% | 80.6% | 72.2% |
| Absence of nearby mountains | 74.8% | 89.3% | 81.6% |
| Presence of nearby mountains | 57.8% | 64.4% | 60.0% |

It is not clear to me how the information of VSI was used to estimate physical variable like ht in eq(10). Did the authors used any hydrological models? If yes, I think it would be appropriate to give a brief description in the methods section.

We did not use any hydrological model because we adopt a model free approach and directly pass the VSI information to the controller. This is due to the fact that a process informed translation of the index into a hydrological model would be extremely complex. The index is extracting information from a localized context and the upscaling to the whole basin would require a physical interpretation of the index, which is beyond the scope of this work. We better clarified this point in the revised manuscript (pag. 11, lines 12-14).

Finally, I believe authors should clearly define the limitations of this study (e.g. computational time of the imagine process and availability of public photo from people) in the conclusion section
Following the referee's suggestion, also pointed out by the second referee, we will add a more balanced discussion about the requirements and limitations of the proposed approach.
The paper represents a proof of concept on the possible use of public media for improving water resources monitoring and management. Our experiment relies on a small portion of the data we crawled and processed. In the revised manuscript, we added a discussion about the main factors which may limit our approach both in terms of computation power and data availability (pag. 18, lines 21-24 and pag. 19, lines 1-7).
For example, the generation of a 1500 px X 12000 px panoramic view requires approximately 1000 ms with a GeForce GTX 850M graphic card. The alignment of an image to the virtual panorama requires approximately 30.000 ms on an OpenStack virtual instance with 4 2.5GHz VCPUs and 8Gb of RAM.
In our case, we split the 300 X 160 km region of the Italian and Switzerland Alps using a 5 X 5 km step grid. We analyzed all the photographs and webcam images acquired in the specified region over a 6 months' period (from December 1st, 2014 to May 31, 2015), for which the availability of photographs and webcams is the following one:
-   Photographs: spatial coverage 38%, temporal frequency ~10.
-   Webcams: spatial coverage 10%, temporal frequency ~10000.
where the spatial coverage is defined as the fraction of grid cells containing at least one image over the considered observation period, while the temporal frequency is defined as the average number of images contained in a non-empty grid cell in the observation period.

**Referee comment #2**

This paper presents an approach to supplement in situ and satellite data in snow dominated watersheds by using publicly available webcam images and flickr photographs. The authors describe a complete procedure from the crawling of the images to the application of the extracted information on the regulation policy of a reservoir lake.
I enjoyed reading this paper and I concur with reviewer 1 that it deserves publication.
We thank the referee for the positive comment.

I am also left with the feeling that the authors may have somehow eluded the limitations of their approach. The discussion should provide a more balanced analysis, e.g. by discussing the computation cost and data storage issues, the minimal amount or frequency of images to reach a stable solution in the VSI, and most importantly the steps that require human intervention (see specific comments marked (A) and (B) below). I spent some time to play around with this type of data so I can imagine the tedious work and the challenges to automatically filter, align and classify webcams or photos.
Following the referee's suggestion, which was also pointed out by the first referee, we will add a more balanced discussion about requirements and limitations of the proposed approach.
As far as the human intervention is concerned, it is worth noting that the requirements of our method are very low. Human intervention is indeed required only for the skyline annotation and the for setting up the experiment on Lake Como basin (e.g., select the webcam to use, ensuring it has enough information). In the revised manuscript, we discuss in detail the main factors currently limiting our approach, especially in terms of its applicability to the entire web media content (pag. 18, lines 21-24 and pag. 19, lines 1-7).

I encourage the authors to distribute an open source implementation of their processing to foster the development of similar applications in other regions.
We are going to release our algorithms as open source implementation. Furthermore, our intent is to transform the web platform into a unique mountain-related media repository, that would provide computer science and environmental researches not only with input data and algorithms, but also with intermediate step results (e.g., somebody interested in testing a new snow pixelwise classification method could start from already aligned and weather-filtered images). We specified this in the revised manuscript (pag. 19, lines 15-17).

I provided below a list of points that should be clarified. I hope that the authors will find my comments useful and look forward to reading an updated version. (NB. the line numbering of the manuscript is awkward, maybe an issue with the Copernicus LaTeX style file)
Specific comments:
P02-L12: AMSR-E derived SWE is generally not considered as "accurate" in mountain regions. Please modify or provide a reference to justify.
The sentence was modified as suggested by the reviewer: *Space-board passive microwave radiometers (e.g., AMSR-E) penetrate clouds but have coarse spatial resolution (25 km).*

P03-L20: I disagree that the assessment of the VSI through the Lake Como experiment is the "only viable evaluation method". There are other validation approaches, including more direct approaches

like a comparison with terrestrial time lapse cameras, comparison with high resolution satellite snow maps, etc. Please clarify or remove this sentence.

*The sentence was modified as suggested by the reviewer: This form of assessment provides an indirect validation of the utility of web and crowdsourced information as the VSI extracted from general-purpose mountain images and the traditional observational data collected with dedicated tools are not comparable directly due to the difference in their physical interpretation and spatio-temporal resolution.*

P05-L19: the skyline is manually defined for a first image. Do you mean that a skyline was manually digitalized on 2000 images (see P05-L09)? If yes this should be more clearly acknowledged. (A)

We are currently running a crowdsourcing experiment for annotating all 2000 skylines as part of our effort to release a public dataset. The experiment described in the paper, instead, relies on a single webcam and required a single skyline annotation. We clarified this aspect in the revised version (pag. 5, lines 22-23).

P05-Eq1: symbols p' and $\tau$ are not defined.

In the equation, p' is a pixel different from p and $\tau$ is a threshold on the Euclidean norm $\| p - p' \|$. We fixed the definition of both variables in the revised manuscript (pag. 5, line 25).

P05-L26: specify what is the edge detection algorithm.

We used the Compass algorithm (Ruzon et al., 2001), an advanced edge detector that uses color distributions. We added this information in the revised manuscript (pag. 5, lines 22-23).
*Ruzon, Mark A., and Carlo Tomasi. "Edge, junction, and corner detection using color distributions." IEEE Transactions on Pattern Analysis and Machine Intelligence 23.11 (2001): 1281-1295.*

P06-L09: why "cross" correlation? I would say correlation only.

We are measuring cross-correlation because we want to quantify not only the similarity between the two edge maps, but the entire set of similarities at every possible position of one w.r.t. another. Correlation alone in this case would be a mere measurement of non-causality of the two edge maps. We clarified this point in the revised manuscript (pag. 6, lines 21-24).

P06-L11: do you define a maximum offset to reduce the computation time, and if yes, how?

We do use a maximum offset of 10 pixels to reduce the computation time (and also to reduce the possible error, since the webcam trembling shifts the image not more than few pixels). The threshold was defined through a trial and error method. We clarified this point in the revised version of the paper (pag. 6, lines 24-25).

P08-L21: this is unclear to me: from the edge images, how do you extract the skyline? If this algorithm works, why was it not applied to the webcam images as well? I foresee many obstacles at this step, like the confusion of cloud edges or snow patches edges with skyline edges.

The skyline is extracted from the edge map with a modified version of the multi-stage graph algorithm by Lie et al. (2005). This was not applied to the webcams as a single annotation was sufficient for obtaining a precise skyline extraction. As the referee correctly pointed out, the algorithm suffered from clouds and challenging meteorological conditions when applied to the usergenerated photographs. To overcome this issue, we are currently working on a Convolutional Neural Network model trained on large sets of images to extract a more robust skyline. We fixed this point in the revised manuscript (pag. 8, lines 25-29).
*Lie, Wen-Nung, et al. "A robust dynamic programming algorithm to extract skyline in images for navigation." Pattern recognition letters 26.2 (2005): 221-230.*

P09-L05: what does "local refinement" mean? do you mean a locally varying transformation of the image? If yes specify the method.
The local refinement step is the application of the same edge-alignment procedure, which is first performed during the global step, with a small max radius (50 pixel) and for each mountain peak independently. This allows the peaks to slightly move in their neighborhood to better adapt to the edges. We will clarify this local refinement step in the revised version of the paper.

P09-L05 (sect 2.3): here I understand that you have used a supervised classification to get the snow mask. Then I suggest to explicit the number of samples and the method to define them. (B)
Yes, we used a supervised classifier trained on a dataset that includes 59 images manually segmented in snow/non-snow areas, ending up with more than 7 million annotated pixels. We clarified this point in the revised manuscript (pag. 9, lines 2-3).

P12-L07 (at the end of the page...): please indicate the number of webcam images and the number of flickr photos that were used for this experiment.
The experiment described in the paper was performed by using the images of a single webcam in Livigno, which ensures a continuous time series of daily images over the time horizon 2013-2014 (see Experiment Setting section). We do expect to obtain better, and more valuable, information by using more webcams along with Flickr photos, where webcams produce a temporally dense series of images of the same view, while crowdsourced photos have better spatial distributions but lower time coverage. Yet, we did not have such data over the period 2013-2014. We mentioned this analysis as a possible future research, which, hopefully, will be possible thanks to the continuous acquisition of new web content through our portal (pag. 19, lines 8-9).

P14-Eq9: define r.
In the equation, r is the daily release from the lake. We added the definition of this variable in the revised manuscript (pag. 14, lines 6-7).

P16-L32: did you try to use the freezing level as an input to the regulation model?
We did not use the freezing level as argument of the operating policy because, in a previous analysis, we run an automatic selection procedure with the Input Variable Selection techniques for identifying which variables are more valuable for informing the lake operations (see the Information Selection and Assessment framework in Giuliani et al. (2015)). The results of this analysis showed that snow-information is more valuable than the freezing level: SWE was always selected as the most informative variable to be considered for improving the baseline solution, while the IVS algorithm never selected the freezing level. This result can be explained by two reasons: 1) the dynamics of freezing level is highly correlated with the seasonality and, therefore, it does not add too much information to the day of the year, which is one of the argument of the baseline policy; 2) the freezing level is independent from the amount of snow stored in the mountains and, therefore, similar values of freezing levels may be associated to the beginning of the

lake inflow peak due to large snow melt as well as to lower inflow if a limited amount of snow was accumulated in the previous months. As a consequence, the freezing level is not able to provide the kind of long lead-time prediction of the volume of water that will be available in the future, which is instead captured by snow-related information.

P18-L05: I created an account and logged in to this website to give it a try but the alignment tool was not really working. The page was not responding when I clicked "continue". It might be a browser issue (I used Firefox 49 on MacOS).
We apologize for this, the problem has been fixed and we invite the referee to try it again.

P19-L09: I am not convinced with the potential of this method in the Atlas mountains because there are few operating webcams and probably a much lower amount of wintertime public photos than in the Alps.
The point is well taken. We removed the reference to the Atlas Mountains and better outlined in the conclusions (pag. 19, lines 2-3 and lines 17-20) the potential limitations of the approach in catchments with few operating webcams and lower number of photos (like Atlas).

[revised manuscript text omitted]